# Caesarean section delivery rates and associated factors in a faith-based referral hospital in Ghana: A retrospective analysis

Ebenezer Jones Amoah[1,2]*, Rita Aklie[3]*, Thomas Hinneh[4], Angela Asare[5], Kinglsey E. Amegah[6]

1 Gomoa Fetteh Kakraba, KAAF University College, Kasoa, Ghana, 2 Ghana Health Service, Jaman North Health Directorate, Sampa, Ghana, 3 Nursing and Midwifery Training College, Pantang, Ghana, 4 School of Nursing, Johns Hopkins University, Baltimore, Maryland, United States of America, 5 Holy Family Hospital, Berekum, Ghana, 6 Department of Data Science and Economic Policy, University of Cape Coast, Cape Coast, Ghana

* ejamoah25@gmail.com (EJA); ritardb17@gmail.com (RA)

## Abstract

### Introduction

The global incidence of caesarean section (CS) deliveries has exceeded the recommended threshold set by the World Health Organization. This development is a matter of public health concern due to the cost involved and the potential health risk to the mother and the neonate. We sought to investigate the prevalence, indications, maternal and neonatal outcomes and determinants of CS in private health facilities in Ghana.

### Method

A retrospective cross-sectional analysis was conducted using data from women who delivered at the Holy Family Hospital from January to February 2020 using descriptive and inferential statistics, with a significance level set at p<0.05.

### Results

The prevalence of CS was 28.70%. The primary indications of C/S include previous C/S, foetal distress, breech presentation, pathological CTG and failed induction. Significant associations were found between CS and breech presentation (AOR = 4.60; 95%CI: 1.22–17.38) p<0.024, previous CS history (AOR = 51.72, 95% CI: 11.59–230.70) p<0.00, and neonates referred to NICU (AOR = 3.67, 95% CI: 2.10–6.42) p<0.00.

### Conclusion

The prevalence of caesarean section (CS) deliveries was higher than the WHO-recommended threshold. Major indications for CS included previous CS, fetal distress, and failed induction. Significant risk factors for CS were previous CS history, breech presentation, and neonates referred to NICU.

**Data Availability Statement:** All relevant data are within the paper and its supporting information files.

**Funding:** The author(s) received no specific funding for this work.

**Competing interests:** The authors have declared that no competing interests exist.

**Abbreviations:** ACOG, American College of Obstetricians and Gynecologists; ANC, Antenatal Clinic; APGAR, Appearance Pulse Grimace Activity and Respiration; CS, Caesarean Section; CPD, Cephalo Pelvic Disproportion; EU, European Union; HB, Hemoglobin; HIV, Human Immune Virus; NICU, Neonatal Intensive Care Unit; SVD, Spontaneous Vaginal Delivery; RDS, Respiratory Distress Syndrome; VBAC, vaginal birth after caesarean; WHO, World Health Organization.

## Introduction

Worldwide, it is estimated that about 213 million women undergo the physiological process of gestation and parturition annually, of which nearly 18.5 million deliveries are performed via the surgical caesarean section (CS) [1]. However, a significant proportion of maternal deaths in low- and middle-income countries occurred among women who had previously undergone a CS [2]. CS a surgical intervention for delivering a baby, necessitates creating an abdominal incision, laparotomy, and a uterine incision, hysterotomy, to extract the fetus from the womb [3]. CS is a surgical technique employed for delivering a baby, where the newborn is extracted through a surgical incision in the lower abdominal region instead of a natural vaginal birth.

CS can be categorized into elective or emergency procedures based on the timing of the operation. Elective C/S involves a planned procedure scheduled during pregnancy to ensure optimal maternal and fetal outcomes through high-quality obstetric care and nursing services. In contrast, emergency C/S procedures are performed in response to unforeseen obstetric complications that threaten both the safety and survival of the mother or the foetus [4, 5].

A recent study has reported a global caesarean delivery rate of 21.1%, with 27.2% and 8.2% rates in developed and least developed countries respectively. About 18.5 million global deliveries are through CS [1]. These rates exceed the 10–15% recommended by the World Health Organization (WHO) [6]. Essentially, more than one in five children are delivered through CS globally [7, 8]. It is projected that 29% of all deliveries will be by CS by 2030 [7].

Since 1990, the CS rate has increased in all regions, with the highest rate observed in Eastern Asia, Western Asia, and Northern Africa, representing 44.9, 34.7, and 31.5 percentage points, respectively [8]. Out of the approximately 38 million CS conducted globally annually, 33.5 million of are done in low- and middle-income countries each year [9, 10]. Relatively, CS rates (7.1%) were low in African region compared with Eastern Asia 63.4% [8].

Like other African countries, the incidence of CS in Ghana varies across according to the level of healthcare rates of 14%, 18.5% and 26.9% [11–14]. CS utilization has both positive and negative consequences. Inappropriate discretion of CS can result in unwanted morbidities and mortalities. Evidence suggests that CS is associated with both short- and long-term risks, which can extend beyond the index delivery and have implications for future pregnancies [15]. A report from the American College of Obstetricians and Gynecologists (ACOG), cited by Gedefaw et al. (2020), indicates that the risk of pregnancy-related morbidity and mortality is significantly higher for women who undergo CS, with rates of 35.9 deaths per 100,000 live deliveries, compared to 9.2 deaths per 100,000 live births for women who have a vaginal delivery [4]. Women who undergo caesarean section are at a higher risk of experiencing maternal mortality and postpartum infections than those who have vaginal deliveries [16]. Inappropriate caesarean section indications do not only pose risks to maternal and neonatal health but increase healthcare costs. On average, caesarean sections are nearly 50% more expensive than vaginal births, adding to the financial burden of healthcare systems [17].

Identifying factors linked to a caesarean section is paramount in deciding whether to perform CS. By recognizing the factors contributing to the high rates of caesarean sections, healthcare providers can implement targeted strategies to prevent and manage complications, ultimately improving maternal and neonatal outcomes. It is also crucial to ensure equitable access to caesarean sections for those who require it while minimizing to ensure judicious use of limited healthcare resources.

Foetal distress, previous caesarean section, non-progress of labour, oligohydramnios, malpresentation, cephalopelvic disorders, hypertensive disorders in pregnancy, preeclampsia, premature rupture of membranes, parity, abnormal lie, failed instrumental delivery, multiple pregnancies are some of the known indications of CS [14, 18–20], However, factors such as

advanced maternal age above 35 years, urban residence, high level of education, poorest wealth index, mulltiparity are some factors that can potentially influence CS rate [21]. The impact of CS on neonatal health has been extensively studied. Infants born via CS may face an increased likelihood of respiratory distress syndrome (RDS), transient tachypnea, and admission to the neonatal intensive care unit (NICU) [22].

While the indication for CS have been extensively studied in public health facilities [11, 13, 23], limited information exists on the incidence and associated factors in private care facilities in Ghana. A study by Aminu (2014) suggests that, despite adequate information on what occurs in public hospitals at the district level, the increasing number of CS performed in private hospitals has not been well documented [24]. Moreover, Gebremedhin (2014) highlights that women who give birth in private institutions are more likely to undergo CS than those who deliver in public facilities [25]. Therefore, there is a need to examine the prevalence and determinants for CS in a private health facility in Ghana. The aim of this study is to examine, specifically the Holy Family Hospital, a faith-based health facility, to guide policymakers in improving maternal healthcare services.

Materials and methods

## Study design

This study employed a retrospective design, utilizing data from the maternal delivery register of women who delivered at the Holy Family Hospital, Ghana from January to February 2020. Non-probability purposive sampling was used.

## Study area

The study was conducted at the obstetrics and gynecology units of faith-based hospital in Ghana, a secondary healthcare facility that offers comprehensive healthcare services. The hospital is equipped with various clinical departments, including the maternity unit, which provides delivery, postnatal, and maternity care services. The maternity unit comprises four delivery rooms overseen by highly experienced health professionals, led by an Obstetrics and Gynecology specialist. The unit's delivery caseload stands at 68%, reflecting its substantial contribution to childbirth services in the hospital. This is a crucial referral hospital serving several health centres, polyclinics, and hospitals within the Berekum East and West, Jaman South and North, Dormaa East, and Central districts, receiving cases requiring advanced or specialized care. As such, it sees a diverse range of cases, from routine deliveries to complicated ones that may require surgical interventions or specialized medical care.

## Sample size

The sample size was based on the total number of women who gave birth at the Holy Family Hospital during January and February 2020, as recorded in the delivery register. Four hundred twenty-nine records were retrieved from the from hospitals' maternal health delivery register. This sample selection was deemed sufficient for the study's objectives, and no additional data were collected beyond what was available in the delivery register.

## Data collection

We developed and piloted a data extraction tool to collect information on maternal demographics, obstetric history, indications for caesarean section, and maternal and neonatal outcomes. Two trained research assistants extracted data from the delivery register under the supervision of the principal investigator. Data for the purpose of research was collected from

the period of 19th to 24th July 2023. Despite having access to potentially personally identifiable information, the authors prioritized privacy and ethical standards. An extensive anonymization process was meticulously implemented, removing all identifiable data. This ensured the data's confidentiality and analytical integrity. The anonymized dataset was the foundation for the research's analysis and findings, demonstrating a commitment to upholding ethical considerations and safeguarding individuals' privacy. The extracted data were entered into a password-protected electronic database.

## Data analysis

The data collected from the delivery register were analyzed using descriptive and inferential statistics. Descriptive statistics was used to summarize the demographic variables of the study population, including age, parity, and gestational age. Categorical variables were presented through the use of frequency and percentage measures. Chi-square tests was use to examine the relationship between categorical variables such as mode of delivery (CS or vaginal delivery) and maternal characteristics such as age, parity, and gestational age. The study utilized logistic regression analysis to identify the factors that were linked to CS. The analysis involved the calculation of odds ratios and their corresponding 95% confidence intervals. The adjustment of the significance level was necessary to account for the potential impact of multiple testing. The statistical significance was determined by considering a p-value of $<0.05$. The statistical analyses were conducted using Stata version 16.0, developed by StataCorp in College Station, TX, USA.

## Ethical considerations

The study obtained ethical approval from the Committee on Human Research, Publications, and Ethics (CHRPE) of Kwame Nkrumah University of Science and Technology (KNUST) (CHRPE/AP/605/23) and the Hospital Management prior to its initiation. Due to its retrospective nature, which involved using existing data from the hospital's delivery register, the requirement for informed consent was deemed unnecessary. To ensure the privacy and confidentiality of patients, a comprehensive anonymization process was implemented, removing personally identifiable information such as names, addresses, and identification numbers from the dataset. These measures were carried out in accordance with established protocols and ethical considerations for retrospective studies utilizing hospital records.

The anonymization procedures aimed to safeguard patient privacy, maintain confidentiality, and comply with ethical guidelines. By rendering the data fully de-identified, the researchers prevented the possibility of re-identifying individual patients. This decision was based on the minimal risk posed to participants and the diligent protection of privacy and confidentiality. Throughout the study, the research team strictly adhered to the ethical guidelines established by the IRB or ethics committee, obtaining all necessary approvals to access and analyze the anonymized data. These rigorous ethical considerations were paramount to ensure the scientific integrity of the study and to uphold the rights and privacy of the patients involved.

## Results

### Demographic characteristics of respondent

A total of 429 deliveries was conducted during the study period. The mean age of the study participants was 28.59 years, with a standard deviation of 6.5. The ages of the participants ranged between 14–45 years. Almost half 196/427 (44.96%) of the study participant who gave

birth in the hospital was between the ages of 21 and 30. About 14% (58/427) of those who gave birth are within the adolescent ages (14–20 years).

The majority, 399 (93.44%) of the study participants, had active health insurance. Also (164/427), 38.41% had education up to JHS level, 26.23% also had education level to SHS, and only 6.79% had no formal education. The majority, 71.43% (305/427), of the mothers resided in urban areas. Concerning their occupation, 28.44% (122/427) majority worked as a trader, 19.81% (85/427) were salary workers (professional) and just a little over 9% (40/427) who delivered at the hospital were students (Table 1).

Table 2 presents the obstetric characteristics of the participants in the study. Concerning the parity of the mothers who delivered, 68.93% (295/429) of mothers reported as multipara, while 31.07% (133/429) were nullipara. Regarding gravidity, 59.58% (255/429) of the mothers have been pregnant 1 to 3 times, whiles 40.42% (173/429) of mothers have been pregnant 4–7 times. The majority 74.76% (314/429), were delivered within the gestational age of 39–43 weeks, and only 3.57% (15/429) were delivered within the gestational age of 28–33 weeks. Regarding the HB of mothers, 11.25% (46/429) were anaemic. 46.72% (185/429) had an ANC attendance of 9–12 visits before delivery, whiles 12.12% (48/429) of mothers only had 1–4 visits before delivery. Concerning PMTCT of HIV, 3.73% (16/429) of mothers were reactive to HIV, and 1.86% (8/429) had no record of HIV testing. Concerning the mode of delivery, the proportion of CS delivery was 28.67% (123/429), whiles 67.83% (291/429) and 3.50% (15/429) were delivered through SVD and Assisted Vaginal Delivery, respectively. Of 123 mothers who delivered through caesarean section, 23.58% (29/123) were elective C/S, while 76.42% (94/123)

**Table 1. Demographic characteristics of respondent.**

| Variables | n = 429 | % |
|---|---|---|
| Mean Age (SD) | | 28.59 (6.5) |
| **Age groups (In years)** | | |
| 14–20 | 58 | 13.58 |
| 21–30 | 192 | 44.96 |
| 31–45 | 177 | 41.45 |
| **Health Insurance** | | |
| No | 28 | 6.56 |
| Yes | 399 | 93.44 |
| **Educational level** | | |
| JHS | 164 | 38.41 |
| No formal education | 29 | 6.79 |
| Primary School | 35 | 8.20 |
| SHS | 112 | 26.23 |
| Tertiary | 87 | 20.37 |
| **Type of Residence** | | |
| Rural | 122 | 28.57 |
| Urban | 305 | 71.43 |
| **Occupation** | | |
| Artisan | 70 | 16.32 |
| Farmer | 61 | 14.22 |
| Professional | 85 | 19.81 |
| Students | 40 | 9.32 |
| Trader | 122 | 28.44 |
| Unemployed | 51 | 11.89 |

**Table 2. Obstetric history of respondent.**

| Variables | n = 429 | % |
|---|---:|---:|
| **Parity of Respondent** | | |
| Nullipara | 133 | 31.07 |
| Multipara | 295 | 68.93 |
| **Gravida of Respondent** | | |
| 1–3 | 255 | 59.58 |
| 4–7 | 173 | 40.42 |
| **Gestation age (In weeks)** | | |
| 28–33 | 15 | 3.57 |
| 34–38 | 91 | 21.67 |
| 39–43 | 314 | 74.76 |
| **Number of Hb checked** | | |
| Anaemia | 46 | 11.25 |
| Normal | 363 | 88.75 |
| **Number ANC visit** | | |
| 1–4 | 48 | 12.12 |
| 5–8 | 163 | 41.16 |
| 9–12 | 185 | 46.72 |
| **HIV status** | | |
| Non- reactive | 405 | 94.41 |
| Reactive | 16 | 3.73 |
| Unknown | 8 | 1.86 |
| **Mode of Delivery is by C/S?** | | |
| No | 306 | 71.33 |
| Yes | 123 | 28.67 |
| **Mode of Delivery type** | | |
| Assisted Vaginal Delivery | 15 | 3.50 |
| C/S | 123 | 28.67 |
| SVD | 291 | 67.83 |
| **C/S type** | | |
| Elective | 29 | 23.58 |
| Emergency | 94 | 76.42 |
| **Maternal Outcome** | | |
| Alive | 428 | 99.77 |
| Dead | 1 | 0.23 |
| **History of Previous C/S** | | |
| No | 398 | 92.77 |
| Yes | 31 | 7.23 |
| **Presentation** | | |
| Transverse Presentation | 3 | 0.7 |
| Cephalic Presentation | 402 | 94.15 |
| Breech Presentation | 14 | 3.28 |

were conducted through emergency caesarean section. Regarding maternal delivery outcomes, 0.23% (1/429) of delivery led to maternal death. 7.23% (31/429) of mothers reported a history of previous caesarean section. Concerning the baby's presentation, 94.15% were cephalic presentation, 3.28% were breech presentation and about 1% were transverse presentation (Table 2).

## Indication for caesarean section

Regarding the indications for caesarean section, the following reasons were identified. Most women had previous C/S (26.56%) and foetal distress (25%). Other reasons include; Pathological CTG (12.5%), Breech presentation (9.38%), Failed Induction (8.59%), Preeclampsia/ Eclampsia (6.25%), Cephalo Pelvic Disproportion (CPD) (5.47%), Twin gestation (4.69%), Postdate (3.13%), Raptured Uterus (2.34%). Also, other indications identified include Ectopic pregnancy, failed internal period version, compound presentation, placenta Abruptio, Abnormal presentation, failed vacuum, prolonged labour and Cord prolapse (Fig 1).

## Neonatal outcomes

Neonatal outcomes following a mother's delivery show that out of 429 deliveries conducted, 227, representing 52.91%, were males, while 47.09% (202/429) were females. The majority, 89.81% (379/429) of the babies, had a normal weight, whiles 10.19% (43/429) had low birth weight. Concerning the APGAR score of the babies in 1 and 5 minutes, the majority, 90.61% (386/429) and 96.25% (411/429), respectively, had a Reassuring score. Out of 429 babies delivered in the hospital within the period, 0.70% (3/429) stillbirths were recorded, of which 1/3 were fresh. 119/429 (27.8%) babies were referred to the Neonatal Intensive Unit (NICU) for further management (Table 3).

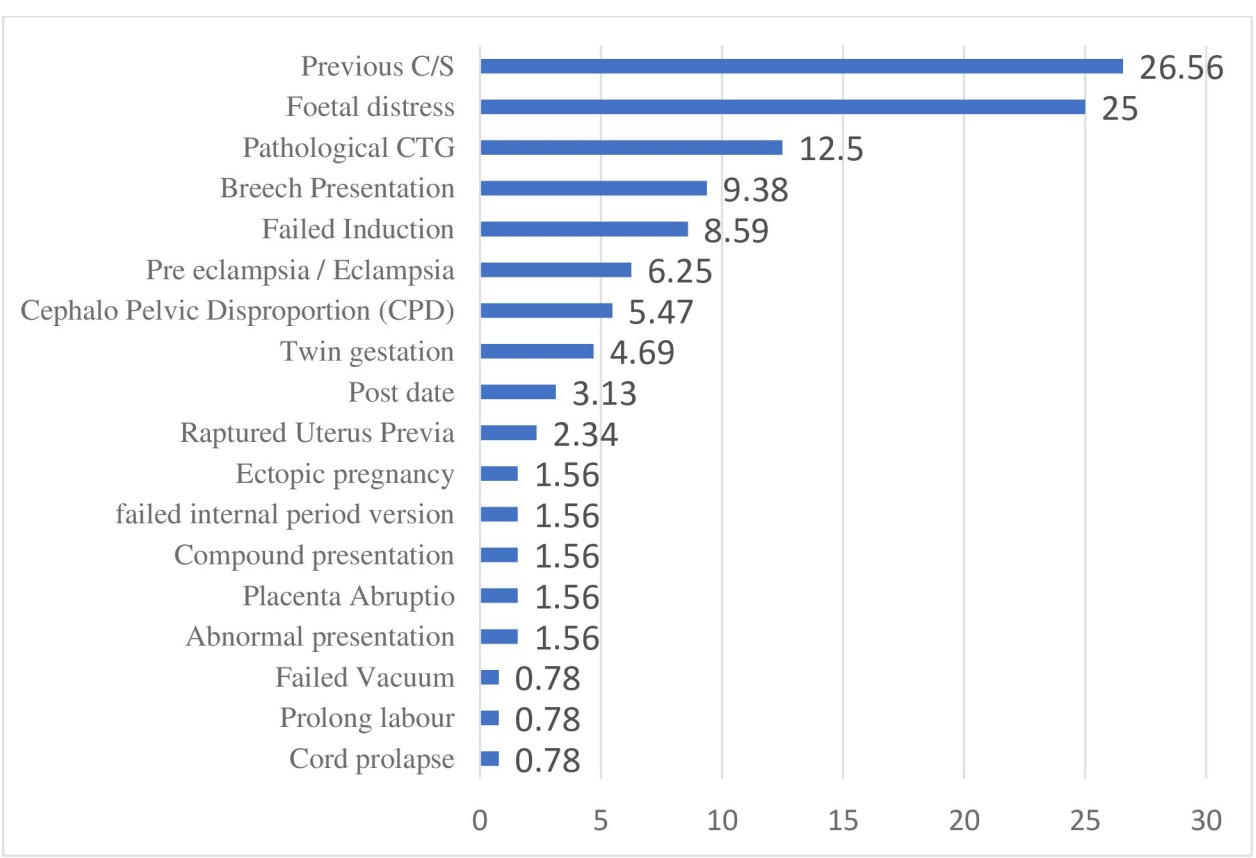

**Fig 1. Indications for caesarean section (multiple response).**

**Table 3. Neonatal outcomes.**

| Variables | n = 429 | % |
|---|---|---|
| **Gender of baby** | | |
| Female | 202 | 47.09 |
| Male | 227 | 52.91 |
| **Baby's Birth Weight (In Kg)** | | |
| low birth weight | 43 | 10.19 |
| Normal | 379 | 89.81 |
| **APGAR score in 1 minute** | | |
| Low | 9 | 2.11 |
| Moderate | 31 | 7.28 |
| Reassuring | 386 | 90.61 |
| **APGAR score in 5 minutes** | | |
| Low | 6 | 1.41 |
| Moderate | 10 | 2.34 |
| Reassuring | 411 | 96.25 |
| **Outcome of baby** | | |
| Dead | 3 | 0.70 |
| Live | 426 | 99.30 |
| **Type of Stillbirth** | | |
| Fresh | 1 | 33.33 |
| Macerated | 2 | 66.67 |
| **Baby referred to NICU** | | |
| No | 309 | 72.20 |
| Yes | 119 | 27.80 |

## Factors associated with caesarean section delivery

The crude and adjusted analyses revealed significant associations between CS delivery and five independent variables: baby's weight, breech presentation, history of previous C/S, gestational age, and neonatal referral to the Neonatal Intensive Care Unit (NICU).

Several noteworthy findings emerged after controlling for other variables at a 95% confidence level in the final multivariable logistic regression analysis. Firstly, women with a fetal breech presentation had a 4.60 times higher likelihood of delivering through caesarean section (AOR = 4.60; 95% CI: 1.22–17.38), p = 0.024. Secondly, women with a history of previous C/S exhibited a significantly increased likelihood of delivering through caesarean section, with an AOR of 51.72 (95% CI: 11.59–230.70), p>0.000 compared to those without a history of caesarean section. Lastly, women whose babies required referral to the NICU for further management were found to have a 3.67 times higher likelihood of undergoing caesarean section (AOR = 3.67; 95% CI: 2.10–6.42), p>0.000 (Table 4).

## Discussion

The aim of this analysis was to estimate the prevalence of caesarean section and associated factors among mothers who gave birth at Holy Family Hospital. Accordingly, the study showed that the prevalence of caesarean section was 28.70%. This is higher than previous studies conducted in Ghana in public health facilities which varies across the various regions [11–14, 26]. The prevalence is higher than the recommended rate of 10–15 percent [6]. Similar studies in different countries confirm these high rates in private health. In Southeastern Brazil, the overall caesarean section rate was the highest rates among women whose deliveries were in a

**Table 4. Factors associated with caesarean section deliveries.**

|  | Crude Odds Ratio | | Adjusted Odds Ratio | | |
| --- | --- | --- | --- | --- | --- |
|  | COR (95% CI) | p-value | AOR (95%CI) | p-value | Sig |
| **Baby's birth weight (In Kg)** | 1 | | 1 | | |
| Normal | 0.42 (0.22–0.80) | 0.009 | 1.61(0.65–4.01) | 0.306 | |
| **Breech presentation** | 6.80 (2.09–22.13) | 0.001 | 4.60 (1.22–17.38) | 0.024 | ** |
| **History of Previous C/S** | 51.22(12.03–218.08) | 0.000 | 51.72(11.59–230.70) | 0.000 | *** |
| **Gestation age (In weeks)** | 1 | | 1 | | |
| 34–38 | 0.72(0.24–2.15) | 0.552 | 0.59(0.15–2.35) | 0.453 | |
| 39–43 | 0.26(0.089–0.729) | 0.011 | 0.41(0.10–1.58) | 0.193 | |
| **Baby referred to NICU** | 3.45 (2.20–5.422) | 0.000 | 3.67 (2.10–6.42) | 0.000 | *** |

*** p < .01

** p < .05

* p < .1

private health facility [27]. Similarly, the caesarean section rate in private hospitals (57.8%) was significantly higher than in public hospitals in Vietnam (49.1%) [28] and likewise in the Republic of Ireland (42.7% vs 25.3%) respectively [29]. One possible explanation for this increase in prevalence is that the hospital serves as a referral centre for the municipality and other adjourning districts, which may contribute to higher cases and a higher prevalence of caesarean section. Referral cases from peripheral health facilities to tertiary hospitals, particularly in low- and middle-income countries, increase caesarean section rates [30, 31].

Other reasons that could be attributed to these high rates are that private health facilities often operate as businesses and rely on profits to remain viable. caesarean sections are typically more expensive than vaginal deliveries, and private health facilities may be more likely to perform a caesarean section as it generates higher revenue. Private health facilities may also have more concerns about legal liability than public health facilities. Even though the prevalence of caesarean section is high in this private facility, there is the contrary of low rates in different countries [32].

According to the study, infants who underwent neonatal intensive care unit (NICU) admission had a significantly higher odds ratio of 3.67 for being delivered by caesarean section. This result is in line with a similar study conducted in the Northern part of Ghana [33]. Moreover, previous studies have consistently reported a significant association between caesarean delivery and an increased likelihood of NICU admission among neonates [34–36]. Indeed, babies referred to the neonatal intensive care unit (NICU) are more likely to have been delivered by caesarean section. Some conditions requiring NICU admission may be more common in babies delivered by caesarean section. For example, babies born via caesarean section may have a higher risk of respiratory distress syndrome (RDS), a condition where the baby's lungs are not fully developed. This can lead to breathing difficulties and require admission to the NICU for oxygen therapy and other treatments. However, babies may be referred to the neonatal intensive care unit (NICU) for various reasons, and not all of them are related to the mode of delivery. While it is true that babies delivered by caesarean section may be more likely to require NICU admission, this is not always the case.

It is possible that the observed increase in the likelihood of caesarean section delivery among infants requiring NICU admission in the study could be attributed to the fact that the hospital in question serves as a referral Centre. This could result in a higher prevalence of complicated deliveries and neonatal conditions, which may lead to a greater likelihood of caesarean delivery and later NICU admission.

Our study revealed that women with previous CS were more likely to undergo another CS delivery, consistent with studies conducted in Ethiopia [37, 38]. Repeat caesarean section may be recommended as part of planned elective delivery for high-risk pregnancies or concerns about the baby's health. Additionally, medical reasons such as placenta previa, where vaginal delivery would be unsafe, may also necessitate a repeat CS. It is important to consider individual circumstances and carefully weigh the potential risks and benefits when deciding on the mode of delivery.

Based on the current study, it was observed that the odds of delivery by CS were higher in women with a fetal presentation of breech. This finding is consistent with previous studies such as one conducted by Ayalew et al. (2020), which reported that the likelihood of caesarean section was 2.5 times higher in mothers with malpresentation than those without malpresentation [38]. A similar study conducted in Felege Hiwot Referral Hospital in the Amhara region of Northwest Ethiopia also found that women with malpresentation were more likely to undergo caesarean section [39]. Breech presentations can pose increased risks during vaginal delivery than when the baby is head-down. Due to the potential complications and risks associated with breech presentation, healthcare providers often recommend a caesarean section as the preferred mode of delivery. This is done to ensure the safety of both the mother and the baby, reducing the likelihood of birth complications and improving overall delivery outcomes.

A study comparing breech delivery to caesarean section found that the risk of neonatal morbidity was twice as high for those who underwent vaginal breech delivery compared to those who had a Caesarean section [40]. This confirms the position of ACOG which recommends caesarean delivery as the preferred mode for breech presentation unless an experienced obstetrician and skilled neonatal care are available for vaginal delivery. Mothers with malpresentation may be unable to give vaginal delivery due to alterations in the normal fetal presentation during labour, leading to the need for delivery via CS.

Furthermore, the findings of our study indicated that the primary indication for caesarean section was women with a history of previous CS. Similarly, a study conducted in Morocco reported that a previous caesarean section was the most common reason for undergoing a Caesarean section [41]. Previous Caesarean section (C/S) increases the likelihood of choosing a CS in subsequent pregnancies due to the risk of uterine rupture during a vaginal birth after a caesarean (VBAC). Uterine rupture is a rare but serious complication where the scar from the previous C/S tears during labour leads to severe bleeding and fetal distress. Many healthcare providers and women prefer a repeat caesarean section to minimize these risks.

Additionally, fetal distress was identified as the second most common indication for caesarean section. This finding aligns with a similar study that reported fetal distress accounted for most caesarean -section deliveries [42]. It is considered an indication for caesarean section (C/S) because it indicates potential harm or distress to the baby and the need for immediate intervention to ensure the baby's well-being. The caesarean section allows for quick and controlled delivery, reducing the risk of further complications and allowing for timely medical interventions to address fetal distress.

Other indications for caesarean section identified in the study include pathological CTG, breech presentation, failed induction, preeclampsia/eclampsia, cephalo-pelvic disproportion, post-term pregnancy, ruptured uterus, placenta abruption, abnormal presentation, failed internal period version, compound presentation, failed vacuum, prolonged labour, and cord prolapse. These indications are consistent with those found in previous studies conducted in Iraq and Tanzania [43–45]. These indications reflect various maternal and fetal factors that may require caesarean section delivery to ensure the mother's and baby's safety and well-being.

## Conclusion

The CS rate reported in this study is higher than the recommended rate of 10–15%. The primary reported indication includes previous C/S, foetal distress, breech presentation, pathological CTG, and filed induction. The occurrence of previous caesarean section, breech presentation, and neonates referred to the neonatal intensive care unit (NICU) were significantly associated with an increased likelihood of caesarean delivery. These findings indicate that the risk factors for caesarean section in this hospital setting are consistent with those reported in previous research and suggest that interventions aimed at reducing the caesarean section rate should target these specific risk factors.

## Limitations

Acknowledging that the prevalence and indications for Cesarean Section (CS) birth have been extensively studied and established, this research adds a unique dimension by centering on a faith-based hospital. Existing studies predominantly focus on public and private health facilities, potentially overlooking distinctive practices and patient profiles in faith-based institutions. The Holy Family Hospital in Ghana, as the focal point of this study, provides valuable insights into CS rates and associated factors within a faith-based setting. By delving into this unexplored territory, the research enriches our understanding of CS prevalence and factors, contributing significantly to the broader landscape of childbirth practices and outcomes across diverse healthcare settings.

However, the findings may not be universally applicable, given the institutional context. Additionally, the study lacks an evaluation of maternal outcomes post-CS, highlighting the need for further research to comprehensively assess complications associated with high CS rates.

## Supporting information

**S1 Dataset.**
(XLS)

## Acknowledgments

The authors are grateful to the Holy Family Hospital, Berekum Management for granting us permission to conduct the study. We are also thankful for the Labour ward staff who were used as research assistants for their cooperation and support.

## Author Contributions

**Conceptualization:** Ebenezer Jones Amoah, Rita Aklie, Kinglsey E. Amegah.

**Data curation:** Ebenezer Jones Amoah, Kinglsey E. Amegah.

**Formal analysis:** Ebenezer Jones Amoah, Angela Asare, Kinglsey E. Amegah.

**Investigation:** Ebenezer Jones Amoah, Thomas Hinneh, Kinglsey E. Amegah.

**Methodology:** Ebenezer Jones Amoah, Rita Aklie.

**Project administration:** Ebenezer Jones Amoah, Thomas Hinneh.

**Resources:** Ebenezer Jones Amoah, Angela Asare.

**Software:** Ebenezer Jones Amoah, Kinglsey E. Amegah.

**Supervision:** Ebenezer Jones Amoah, Rita Aklie, Thomas Hinneh, Angela Asare.

**Validation:** Ebenezer Jones Amoah, Rita Aklie, Thomas Hinneh, Angela Asare, Kinglsey E. Amegah.

**Visualization:** Ebenezer Jones Amoah, Angela Asare.

**Writing – original draft:** Ebenezer Jones Amoah, Rita Aklie, Thomas Hinneh, Angela Asare.

**Writing – review & editing:** Ebenezer Jones Amoah, Rita Aklie, Thomas Hinneh.

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
