## [Decision Letter · Decision Letter 0]

24 Jan 2024

PONE-D-23-29508Caesarean section delivery rates and associated factors in a faith-based referral Hospital in Ghana: A retrospective analysis.PLOS ONE

Dear Dr. Amoah,

Thank you for submitting your manuscript to PLOS ONE. After careful consideration, we feel that it has merit but does not fully meet PLOS ONE’s publication criteria as it currently stands. Therefore, we invite you to submit a revised version of the manuscript that addresses the points raised during the review process.

Thank you for submitting your manuscript. Both reviewers have provided detailed feedback and have suggested minor revisions. Therefore, I invite you to respond to the reviewers' comments and submit your revised manuscript.

We look forward to receiving your revised manuscript.

Kind regards,

Sunita Panda, PhD

Academic Editor

PLOS ONE

Journal Requirements:

4. Please upload a copy of Supporting Information Figure/Table/etc. S1 Fig1 and S2 Fig2 which you refer to in your text on page 20.

Reviewers' comments:

Reviewer's Responses to Questions

**Comments to the Author**

1. Is the manuscript technically sound, and do the data support the conclusions?

Reviewer #1: Yes

Reviewer #2: Yes

2. Has the statistical analysis been performed appropriately and rigorously? 

Reviewer #1: Yes

Reviewer #2: Yes

3. Have the authors made all data underlying the findings in their manuscript fully available?

Reviewer #1: Yes

Reviewer #2: Yes

4. Is the manuscript presented in an intelligible fashion and written in standard English?

Reviewer #1: Yes

Reviewer #2: Yes

5. Review Comments to the Author

Reviewer #1: 1. The prevalence and indications for undergoing CS birth have already been explored in many studies and established. There is a lack of novelty in this study.

2. The study design may be written as a retrospective study design instead of a cross-sectional retrospective.

3. correct as Nonprobability purposive sampling,- not nonprobable sampling

4. The study setting must specify the delivery rate and delivery load with labour facilities instead of giving other general information.

5. The sample size calculation was not mentioned in the study.

6.Table information to be corrected as n, % instead of Freq. Percent

7.why only Neonatal Outcomes are collected, and moreover no such objectives are mentioned related to this information?

8.Indications for referral of baby to NICU-also not a part of your study objective

Reviewer #2: Dear Authors,

Thankyou for your submission.

The manuscript is simple and easy to understand. Discussion chapter is well handled. However, there are few grammatical errors that need revision and few concerns that needs to be addressed.

1. Kindly ensure the spelling of a particular word remains the same throughout. for e.g. (Caesarean)there are two different

spellings used.

2. (36)in abstract should be failed induction instead of filed.

3. (172-176) grammar- tense need to be corrected. Kindly ensure the explanation is in one particular tense( there is a mix

of past and present tense).

3. (185-194) statistical data needs revision. Follow one method, either roundup the numbers or place the details in decimal

s throughout.

4. (189) the stats mentioned here don't match the one projected in the table.

5. (200-201) Statistics needs revision

6. (210) should be ruptured uterus

7. (250) table doesn't reflect all the factors

8. Regarding the comprehensive anonymization process for removing personally identifiable information

such as names, addresses, and identification numbers from the dataset, is there a established protocol?. if yes, who is

involved in segregating and hiding the identity?. Is the person aware of the study to be undertaken?

9. Were birth injuries in newborn taken into consideration?

6. PLOS authors have the option to publish the peer review history of their article (what does this mean?). If published, this will include your full peer review and any attached files.

Reviewer #1: **Yes: **Dharitri Swain

Reviewer #2: No

---

## [Author Response · Author response to Decision Letter 0]

27 Jan 2024

Journal Submissions Rebuttal Letter

Sunita Panda, PhD

Academic Editor

PLOS ONE

27th January 2024.

Dear Dr Panda,

Re: Resubmission of a manuscript (PONE-D-23-29508)

Thank you for inviting us to submit a revised draft of our manuscript entitled, "Caesarean section delivery rates and associated factors in a faith-based referral Hospital in Ghana: A retrospective analysis” PLOS ONE for publication. We also appreciate the time and effort you and other reviewers have dedicated to providing insightful feedback on ways to strengthen our paper. Thus, it is with great pleasure that we resubmit our manuscript for further consideration. We have incorporated changes that reflect the detailed suggestions you have graciously provided. We also hope that our edits and responses below satisfactorily address all the issues and concerns you have noted.

Below we provide the point-by-point responses. All modifications in the manuscript have been highlighted in red and are in track changes.

Yours Sincerely,

Ebenezer Jones Amoah (corresponding author)

ejamoah25@gmail.com

Reviewer #1: 

Comment 1: The prevalence and indications for undergoing CS birth have already been explored in many studies and established. There is a lack of novelty in this study.

Responses: Thank you for the observation. Acknowledging that the prevalence and indications for Cesarean Section (CS) birth have been extensively studied and established, this research adds a unique dimension by centering on a faith-based hospital. Existing studies predominantly focus on public and private health facilities, potentially overlooking distinctive practices and patient profiles in faith-based institutions. The Holy Family Hospital in Ghana, as the focal point of this study, provides valuable insights into CS rates and associated factors within a faith-based setting. By delving into this unexplored territory, the research enriches our understanding of CS prevalence and factors, contributing significantly to the broader landscape of childbirth practices and outcomes across diverse healthcare settings. See line 364 - 382

Comment 2: The study design may be written as a retrospective study design instead of a cross-sectional retrospective.

Response: Thank you once again for the observation. the study design has been written as "this study employed a retrospective design"

Comment 3: correct as Nonprobability purposive sampling,- not nonprobable sampling

Response: thank you and has been corrected as "Nonprobability purposive sampling"

Comment 4: The study setting must specify the delivery rate and delivery load with labour facilities instead of giving other general information.

Response: Thank you for the observation. delivery caseload and labour facilities has been added.

Comment 5: The sample size calculation was not mentioned in the study.

Response: Thank you. The study design, being retrospective, did not explicitly involve a formal sample size calculation. Instead, it adopted a census approach, encompassing all women who gave birth at Holy Family Hospital during January and February 2020, as recorded in the delivery register. Four hundred twenty-nine records, constituting the entire population for the specified period, were retrieved from the hospital's maternal health delivery register. This comprehensive sampling was considered adequate for meeting the study's objectives, and no supplementary data were collected beyond the available records in the delivery register.

Comment 6: Table information to be corrected as n, % instead of Freq. Percent

Response: very appreciative. Table information corrected as n, %

Comment 7: why only Neonatal Outcomes are collected, and moreover no such objectives are mentioned related to this information?

Response: Thank you and grateful for the observation. Neonatal outcome is one of the objectives, related information have been provided in the introduction.

Comment 8: Indications for referral of baby to NICU-also not a part of your study objective

Response: Indications for referral of baby to NICU was not part of the objectives and have subsequently been removed.

Reviewer 2:

Comment 1: Kindly ensure the spelling of a particular word remains the same throughout. for e.g. (Caesarean)there are two different spellings used.

Response: All spelling caesarean and cesarean have been change to caesarean 

Comment 2: (36)in abstract should be failed induction instead of filed.

Response: Thank you for the observation on Line 36 : filed induction in abstract change to change failed induction

Comment 3: (172-176) grammar- tense need to be corrected. Kindly ensure the explanation is in one particular tense( there is a mix of past and present tense).

Response: Thank you for the observation once again. Please various tenses has been corrected.

Comment 4. (185-194) statistical data needs revision. Follow one method, either roundup the numbers or place the details in decimals throughout. 

Response: Thank you sir on Line 185-194: Statistical data revised to decimals throughout

Comment 5: (189) the stats mentioned here don't match the one projected in the table.

Response: Thank you. Line 189: statistical data revised to match with one in projected table

Comment 5: (200-201) Statistics needs revision

Response: Line 200-201: statistical data revised. Thank you

Comment 6: (210) should be ruptured uterus

Response: Line 210: raptured uterus previa change to raptured uterus. Very grateful for the observation.

Comment 7: (250) table doesn't reflect all the factors

Response: Thank you for the enquiryon Line 250: table on factors . The table selectively includes factors that exhibited a statistically significant association or influence on Caesarean section (CS), excluding those that did not contribute significantly to the predictive model. This approach aims to offer a focused representation of key determinants impacting the likelihood of CS based on statistical significance.

Comment 8: Regarding the comprehensive anonymization process for removing personally identifiable information such as names, addresses, and identification numbers from the dataset, is there a established protocol?. if yes, who is involved in segregating and hiding the identity?. Is the person aware of the study to be undertaken?

Response: Thank you for raising an important point regarding the anonymization process in our study.

Following the explicit recommendation from the hospital management, we diligently adhered to their guidelines to implement a robust anonymization process. This process was meticulously executed by our dedicated data management and research team, all of whom have undergone comprehensive training in ethical considerations and privacy protection.

In alignment with the hospital's directives, the anonymization protocol involved the careful removal of personally identifiable information (PII), including names, addresses, and identification numbers from the dataset. Our team, well-versed in the study's objectives, ensured that this crucial step was performed with the utmost care to safeguard the privacy and confidentiality of the participants.

It's important to emphasize that, in accordance with ethical standards, researchers were briefed on the anonymization process. However, the actual identities of individuals were not disclosed to researchers, reinforcing our commitment to maintaining the highest level of confidentiality and protecting participant privacy.

We appreciate your attention to this aspect of our study and would like to reassure you that we have taken every precaution to uphold ethical standards in data management as per the hospital management's recommendation.

Comment 9: Were birth injuries in newborn taken into consideration?

Response: Birth injuries in newborns were not considered in the study as there was no available data on this specific aspect.

CONCLUDING REMARKS: Again, thank you for allowing us to strengthen our manuscript with your valuable comments and queries. We have worked hard to incorporate your feedback and hope that these revisions persuade you to accept our submission.

---

## [Decision Letter · Decision Letter 1]

20 Mar 2024

Caesarean section delivery rates and associated factors in a faith-based referral Hospital in Ghana: A retrospective analysis.

PONE-D-23-29508R1

Dear Dr. Amoah,

We’re pleased to inform you that your manuscript has been judged scientifically suitable for publication and will be formally accepted for publication once it meets all outstanding technical requirements.

Kind regards,

Sunita Panda, PhD

Academic Editor

PLOS ONE

Additional Editor Comments (optional):

Reviewers' comments:

Reviewer's Responses to Questions

**Comments to the Author**

1. If the authors have adequately addressed your comments raised in a previous round of review and you feel that this manuscript is now acceptable for publication, you may indicate that here to bypass the “Comments to the Author” section, enter your conflict of interest statement in the “Confidential to Editor” section, and submit your "Accept" recommendation.

Reviewer #1: All comments have been addressed

Reviewer #2: All comments have been addressed

2. Is the manuscript technically sound, and do the data support the conclusions?

Reviewer #1: Yes

Reviewer #2: Yes

3. Has the statistical analysis been performed appropriately and rigorously? 

Reviewer #1: Yes

Reviewer #2: Yes

4. Have the authors made all data underlying the findings in their manuscript fully available?

Reviewer #1: Yes

Reviewer #2: Yes

5. Is the manuscript presented in an intelligible fashion and written in standard English?

Reviewer #1: Yes

Reviewer #2: Yes

6. Review Comments to the Author

Reviewer #1: (No Response)

Reviewer #2: Dear Authors,

I am glad to receive the corrections on the stated observations. All the comments made have been constructively revised. Though there is not much novelty in the subject, it has definitely created a database for specific area that lacked the information and for which the study was intended.

7. PLOS authors have the option to publish the peer review history of their article (what does this mean?). If published, this will include your full peer review and any attached files.

Reviewer #1: **Yes: **Dharitri Swain

Reviewer #2: **Yes: **Prof. Delphina Mahesh Gurav

---

## [Editor Report · Acceptance letter]

3 Apr 2024

PONE-D-23-29508R1 

PLOS ONE

Dear Dr. Amoah, 

I'm pleased to inform you that your manuscript has been deemed suitable for publication in PLOS ONE. Congratulations! Your manuscript is now being handed over to our production team.

Kind regards, 

on behalf of

Dr Sunita Panda 

Academic Editor

PLOS ONE